# Recycling of Cement Industry Waste for Alkali-Activated Materials Production

**DOI:** 10.3390/ma15196660

**Published:** 2022-09-26

**Authors:** Madina Salamanova, Sayd-Alvi Murtazaev, Magomed Saidumov, Arbi Alaskhanov, Tamara Murtazaeva, Roman Fediuk

**Affiliations:** 1Grozny State Oil Technical University named after acad. M.D. Millionshchikov, 364051 Grozny, Russia; 2Polytechnical Institute, Far Eastern Federal University, 690922 Vladivostok, Russia; 3Peter the Great St. Petersburg Polytechnic University, 195251 St. Petersburg, Russia

**Keywords:** eco-friendly, binder, aspiration dust, clinker dust, alkaline activation, zeolite phase

## Abstract

The cement industry is recognized as an environmental nuisance, and so there is a need to not only minimizes the consumption of cement, but also to completely recycle the waste of the cement industry. This paper’s originality lies in the fact that, for the first time, a comprehensive study of the structure formation of alkali-activated materials (AAM) based on aspiration dust and clinker dust has been carried out. The tasks for achieving this goal were to characterize cement production waste as a new binder and comprehensively research the microstructure, fresh, physical, and mechanical properties of alkali-activated material based on a cement-free binder. Grains of cement production waste are represented by coarse volumetric particles with pronounced cleavage, and a clear presence of minerals is observed. The mineral composition of cement production waste is characterized by calcium silicates, which guarantee good binding properties. The results of the X-ray diffraction analysis of the samples (based on the alkaline-activated cement-free binder using clinker dust and aspiration dust) confirmed the presence of calcite, quartz, feldspar close to albite, micas, and zeolites. The obtained products of the chemical interaction of the binder components confirm the effectiveness of the newly developed AAM. As a result of comparing several binders, it was found that the binder based on aspiration dust with Na_2_SiO_3_ and Na_2_SiF_6_ was the most effective, since, for specimens based on it, a density of 1.8 g/cm^3^, maximum compressive strength of 50.7 MPa, flexural strength of 5.6 MPa, minimum setting time (starting at 24 min and ending at 36 min), and water absorption of 12.8 wt. % were obtained. The research results will be of interest to specialists in the construction industry since the proposed recipes for eco-friendly, alkali-activated materials are an alternative to expensive and energy-intensive Portland cement, and they provide for the creation of strong and durable concrete and reinforced concrete composites.

## 1. Introduction

Portland cement production at the world level in 2021 reached 4.1 billion tons, and it is growing rapidly from year to year due to developing countries [1]. The global cement industry occupies one of the leading places (after the electric power and transport industries) in terms of greenhouse gas emissions (responsible for approximately 10%) [2]. A huge amount of carbon dioxide is released into the air, which has been preserved in the rocks and minerals of various genesis for billions of years, which ultimately affects the ecological situation of the troposphere [3]. In addition, clinker dust settles on the roofs of industrial buildings and then hydrates, forming buildups that can lead to the collapse of structures. Part of this dust (aspiration) is captured by filters and needs to be further disposed of. Therefore, there is a need to minimize the consumption of cement with simultaneous utilization in the production of building materials.

There is a need to search for cementless building materials, the first place among which is occupied by alkali-activated materials (AAM) [4]. It was established that AAM, having high strength characteristics, are distinguished by their improved performances and structural parameters, such as denser structures, low solubilities of hydration products, and low porosities, which, in combination, increase water resistance and frost resistance [5,6,7]. Burduhos Negris et al. [8] comprehensively reviewed geopolymers and their uses. Yang et al. [9] proved the high corrosion resistance of slag-alkaline concretes against magnesia and sulfate aggressiond in low-mineralized environments with hydrocarbonate hardness. Mohseni [10] developed low-exothermic slag-alkali concretes with low heat release, which allowed the use of these developments in monolithic concreting and, in particular, in the construction of massive elements of buildings and structures. Ozturk [11] confirmed the effectiveness of using slag-alkaline concretes at low temperatures, which is explained by the low freezing temperature of the mix. Mahdi et al. [12] established the high strength of slag-alkali materials and the possibility of condensation-crystallization hardening, which is due to cation exchange between alkali metal salts of the R_2_CO_3_ type and the calcium oxide of granular slag, resulting in the formation of calcium carbonate, sodium hydroxide, and potassium solutions.

The traditional materials for creating alkali-activated materials include fly ash and blast-furnace granulated slag [13,14]. There is also a constant search for a new raw material base for these composites [15]. Mahdi et al. [12] investigated the strength and durability properties of geopolymer paver blocks made with fly ash and brick kiln rice husk ash. Zhu et al. [16] researched the effects of carbon nanofibers on the hydration and geopolymerization of low- and high-calcium geopolymers. Numerous studies in recent years have been devoted to the joint and separate use of fly ash and rice husk ash in alkali-activated materials [17,18,19,20].Various methods of activating binders for alkali-activated materials are being developed and constantly improved [21]. Yavuz et al. [22] characterized class C and F fly ashes based geopolymers that incorporated silica fume. Shoaei et al. [23] investigated glass powder as a partial precursor in Portland cement and alkali-activated slag mortar. Good results have been obtained with the integrated use of waste from thermal power plants such fly ash and coal ash in the production of alkali-activated materials [24]. Volodchenko et al. [25] improved its composite performance based on non-conventional natural and technogenic raw materials. Paya et al. [26] studied the application of alkali-activated industrial waste. Gao et al. [27] comprehensively investigated the behavior of metakaolin-based potassium geopolymers in acidic solutions.

Bernal and Provis [28] argued that an expansion of the raw material base for the production of geopolymers is needed, which will increase their durability. There are few studies on the use of cement industry waste as a precursor for the production of alkali-activated materials [29,30]. However, in studying the processes of AAM structure formation, for example, with fly ash, a good prospect for the use of clinker dust and aspiration dust for these purposes is revealed [31,32]. Danish et al. [33] suggested that the use of cement waste in the production of materials with different types of hardening can provide some degree of self-healing properties in service. The same authors, developing this idea in another paper [34], recommended a more comprehensive study and use of clinker dust, including clinker dust together with other production wastes (for example, cenospheres). Thus, in the present time, activities at both the industrial and experimental levels are focusing on the implementation of alkali-activated composites, and they are still underway with the use of mineral fine powders with a chemical composition identical to the clinker dust of man-made origins. Therefore, the study of the structure formation of alkali-activated materials using cement production wastes as precursors is very promising. 

Rotary clinker kilns are the main sources of dust emissions (30% of dust removal), and this is due to the fact that the clinker firing process occurs according to the counterflow principle where exhaust flue gases entrain fine particles of the fired raw mixture into their flow [35]. In the cold end of a clinker kiln, dust is released (called aspiration dust), and its composition is similar to that of the initial raw mixture [36]. Clinker dust is generated in the last hot zone of the kiln, in the cooling sections, and in the clinker conveyor gallery [37]. It has been established that the production of each kilogram of clinker accounts for up to 7.5 m^3^ of exhaust gases, with a dust content of up to 70 g/m^3^ [38]. Thus, in order to prevent harmful emissions into the atmosphere and to prevent harm to the natural environment and humanity, furnace gases must be carefully dedusted when passing through the purification system [39]. An electrostatic precipitator can be considered an effective dust collector as its degree of purification is 98–99%, with a dust content at the inlet of 25–30% and an allowable dust concentration in emissions of 0.1–0.5 g/m^3^ [40].

In the process of burning Portland cement clinker in electrostatic precipitators and rotary kilns, a large amount of dust accumulates—both clinker and aspiration. The rational use of these products, which contain a certain proportion of a full-fledged raw material resources, is an urgent task for the cement industry [41]. It should be noted that the reuse of electrostatic precipitator dust by returning it to the kiln is impossible since it contains a certain amount of alkali-containing impurities, and this will negatively affect the quality of the clinker [42].

This paper presents the results of an examination of cement industry waste since their volumes are substantial enough to provide for the rational use of a dust from electrostatic precipitators as a relevant solution to many problems comprising both economical and eco-technical issues. The aim of this work is a to complete comprehensive study of the structure formation of alkaline activated materials based on clinker dust and aspiration dust. The tasks to achieve this aim were the characterization of the cement production waste as a new binder and a comprehensive study of the microstructure, fresh, physical, and mechanical properties of alkali-activated material based on the binder.

## 2. Materials and Methods

### 2.1. Materials

Aspiration and clinker dust from the electrostatic precipitators of rotary clinker kilns (Chechencement, Chiti-Yurt, Russia) were used as a precursor of AAM. Table 1 lists the dusts’ physical properties. Clinker dust is a dark gray powder which is rather abrasive. The fineness of grinding, determined by sieving using a No. 008 sieve, showed a residue of 23%. Aspiration dust is a light beige powder which is much more dispersed than clinker dust, and its fineness of grinding was 18%.

Figure 1 presents the dusts’ appearance and Table 2 lists chemical composition of the dusts.

Liquid glass such sodium metasilicate Na_2_SiO_3_ was used as the binder modifier (silicate modulus of 2.8 and density of 1.42 g/cm^3^). Sodium hydroxide NaOH was used as an alkali. To accelerate the hardening process of the AAM, sodium fluorosilicon Na_2_SiF_6_ was used in a dosage of 6% of the Na_2_SiO_3_ mass.

The quartz sand obtained by fractionation of fine (1.5 mm) and coarse (2.5 mm) grains in a ratio of 22: 78% was used as a fine aggregate. 

### 2.2. Mix Design

Five different concrete mixes were developed; common to all was the use of cement industry waste (Table 3). Two mixes used clinker dust (K1 modified with Na_2_SiO_3_ and K2 with water to prove the effectiveness of the precursor). The other three mixes used aspiration dust (A1 modified with Na_2_SiO_3_, A2 with Na_2_SiO_3_ and the hardening accelerator Na_2_SiF_6_, and A3 mixed with water). 

The prepared samples hardened on the first day under normal conditions at a temperature of 20 ± 2 °C, but on the second day, the samples were placed periodically, for 28 days, in an oven at a temperature of 50 °C for two hours. Samples prepared in this way, even at a high standard consistency, will effectively structure without cracking.

### 2.3. Methods

The granulometry of the raw material particles was carried out using the laser analyzer Analysette 22 (Fritsch, Iber-Oberstein, Germany). The specific surfaces of the bulk raw materials were studied using a PSH-12 device (Khodakov Devices, Moscow, Russia).

The structure formation processes of the AAM were studied using the scanning electron microscope (SEM) Vega II LMU (Tescan, Brno, Czech Republic) with an energy dispersive microanalysis system (Inca energy 450/XT (Silicon Drift detector (ADD; resolution 133 eV at a count rate of 20,000 pulses/s))) manufactured by the company Oxford Instruments Analytical (Oxford, UK). The system provided the ability to carry out elemental analyses in the range from Na to U (lighter elements were not determined, and oxygen was calculated by stoichiometry). The studies were carried out at an accelerating voltage of 20 kV.

X-ray diffraction (XRD) analysis was performed for reflection according to Bragg–Brentano by an ARLX’TRA (Waltham, Massachusetts, U.S.) diffractometer using the Θ–Θ kinematic scheme with a horizontal arrangement of a flat sample. The characteristic radiation of a copper anode was used (wavelengths CuKα1 1.5406 Å and CuKα2 1.5444 Å). The energy window of the semiconductor detector tuned to register this range also partially captured the close wavelengths of CuKβ1 1.3922 Å and WLα1 1.4763 Å.

Studies of the fresh and hardened properties of the AAM were carried out in accordance with Russian standard 30744-2001. Compressive strength was determined on 70 × 70 × 70 mm samples, but for flexural strength, the tests used sample beams with a size of 40 × 40 × 160 mm (three samples of each composition) (Figure 2). 

## 3. Results and Discussion

### 3.1. Characterization of the Dusts

Energy-dispersive microanalysis of the studied electrostatic precipitator dust powders (Figure 3) showed the similarities in the chemical compositions of the clinker dust (a) and aspiration dust (b), respectively, with Portland cement clinker and the initial raw mixture. However, it should be noted that the amount of alkali metal oxide K_2_O in the aspiration dust sample was 6.43%, while in the clinker dust was 1.57%. The explanation for this is that the clinker dust was formed in the hot zone of solid-phase synthesis in a rotary kiln at temperatures of 1300 °C and above, which is where the burnout and decomposition of alkali metal oxides occur.

The SEM images in Figure 3 and Figure 4 present a comparative analysis of the electrostatic precipitators’ dust structures. Grains of clinker dust at a magnification of 5000 times are represented by coarse volumetric particles with pronounced cleavage, and the clear presence of minerals is observed (Figure 4). The microstructure of clinker dust grains is represented by alite and belite crystals and a fine-grained intermediate phase, and there are separate rounded CaO crystals, angular MgO, and pores. The alite crystals are characterized by the appearance of hexagonal or rectangular prisms, while the belite crystals have jagged edges and a rounded shape. Between the alite and belite crystals is an intermediate substance belonging to calcium aluminoferrites, tricalcium aluminate, and clinker glass. The pores look like “wells” of irregular shape and are clogged with iron oxide.

The SEM images of the aspiration dust denote a more loose, porous grain morphology with the observed initial stage of crystallization (Figure 5). Analysis of the SEM images of aspiration dust showed that the grains have a looser, porous structure, characteristic of the initial stage of mineral formation. The particles of aspiration dust are characterized by a small, specific surface and, they practically do not differ in the nature of the distribution of particles. There is an uneven distribution of large aggregates covered with a fine substance of various morphological structures and dimensions, and fine-crystalline microparticles of quartz and calcite with a characteristic cleavage on the surface of the solid phase are clearly identified. Instead of angular components, rounded particles are observed, with no clear interfaces between them.

### 3.2. Microstructure of Alkali-Activated Material

The mix with aspiration dust is characterized by a bulk inhomogeneous structure (Figure 6). Cavities, microcracks and intracrystalline spaces contain developed needle crystals (of up to 200–300 µm in length) of calcium sulfoaluminates (Figure 6a), which occasionally demonstrate intergrowths in the form of “tomentous” clusters, while fine fiber gypsum is present in combination with sulfoaluminates. Lamellar crystals of calcium hydroaluminates are also often detected in cavities (Figure 6b). The chemical composition varies in the clusters with different structural and textural features (Figure 6c,d), and there are bulky crypto-crystalline clusters with an aluminosilicate composition together with the major fraction of decrystallized fractured clusters composed of hydrated calcium silicates.

According to the results of the X-ray diffraction analysis of the AAM produced with the aspiration dust, calcite, quartz, larnite, feldspars with various compositions, and magnesium oxide were present, and weak X-ray peaks corresponding to muscovite (2 Θ ~ 8.8–8.9 degrees) and zeolites were detected (Figure 7). The highest zeolite peak corresponded to garronite, though there was a very weak fluctuation in the region of the analcime-related peaks, which did not exclude its potential presence in scarce amounts. Zeolite reflections are close to garronite or Na-phillipsite and the intensity of the reflections is low, which is explained by their low crystallinity.

The electron microscopy microanalysis denoted the prevailing content of calcium silicates and their hydrates within the main microcrystalline volume. The results of the analysis of the crystal aggregates of the calcium silicates are shown in Table 4 and Figure 6c,d.

The analysis of the results obtained in this study confirmed that alkali cement bricks produced with aspiration dust contained the following minerals: calcite, quartz, larnite, feldspars with various compositions, magnesium oxide, muscovite and zeolites, hydrosilicates, and calcium hydroaluminates and hydrosulfoaluminates. 

The analysis of the alkali-activated materials produced with the clinker dust shows that the samples were characterized by a bulk, relatively uniform structure (Figure 8) formed by the hydrides of calcium silicates (1) and aluminosilicates (2).

According to the results of X-ray diffraction analysis of the AAMs produced with the clinker dust, calcite, quartz, phase related to larnite, and plagioclase related to albite were present. In addition, there was a wide peak in the region of 2 Θ ~ 8.8°–9.1°, which denoted the presence of mica with various structural parameters, and some peaks corresponded to zeolite related to garrronite (Figure 9). The microstructure of the groundmass was formed by non-crystallized clusters of a hydroaluminosilicate “zeolite” composition with a variable Ca/Na ratio, calcite, and phases close in composition to dicalcium silicate hydrates and calcium hydroxide, as well as iron and magnesium, which are associated with aluminosilicate hydrate compounds.

The analysis of the oxide composition of the typical phases denoted that it could be attributed to the phase related to larnite and calcium silicates hydrates (Table 5).

The analysis of the results obtained in this study confirmed that alkali cement bricks produced with the clinker dust contained the following minerals: calcite, larnite, muscovite, zeolites, hydrosilicates, and calcium hydroaluminates and hydrosulfoaluminates.

### 3.3. Fresh, Physical, and Mechanical Properties of the Alkali-Activated Material

The standard consistency of the AAMs based on the aspiration dust was characterized by a fairly high demand for an alkaline solution of 72.5%, and the setting time was rather short, beginning at 16 and ending at 31 min. Even when mixing with water, both dusts (clinker and aspiration) exhibited binder properties, with setting occurring in 6–7 h and compressive strengths of 7.8–7.9 MPa (Table 6). 

The composition of the aspiration dust from the electrostatic precipitators was similar to the prepared raw material mixture, though it contained up to 7% alkalis. To obtain a slurry with a normal density, 72.5% of the alkali solution was consumed, which is a relatively high value for its low specific surface area (S_SP_ = 210 m^2^/kg). The high consumption of the mixing fluid was due to the dust from the electrostatic precipitators consisting of weakly burned clay minerals, not decomposed calcite, since the temperature at the waste collection point within the kiln was 170–320 °C.

The composition of the clinker dust collected from hot end of the kiln for the clinker synthesis matched the composition of the prepared Portland cement clinker, with an alkalis content of up to 2% due to the viscosity of the alkali mixing fluid. As such, the preparation of the slurry with normal density required up to 50% of the mixture.

When activated with an alkali solution, the aspiration dust demonstrated good reactivity, wherein the start of the setting process was observed within 16 min and ended at 31 min. The dust–alkali slurry was quite light and plastic, and easily treatable. In order to compare and reveal the optimum formulation based on the aspiration dust, it was mixed with different mixing fluids with the use of a curing accelerator, in one instance. At the same time, in the system “aspiration dust–Na_2_SiO_3_–Na_2_SiF_6_”, we found that the accelerator caused the opposite effect of the false setting, wherein the setting time slowed down by 8 min, which confirmed that the main binder in this system was the mineral powder. The aspiration dust mixed with water demonstrated a high water loss in the first minutes of study, while setting began quite late (after 6 h). It appeared that it was inactive in its natural form, and an alkali solution is desirable to activate this powder.

The clinker dust from the electrostatic precipitators manifested itself in quite a different way. When mixed with sodium silicate solutes, 50% of the alkali solution was required to obtain a slurry with a standard consistency, though when mixed with water, it demonstrated a hydrophobic effect, and 30% of the water was required to obtain a slurry with a normal density. At the same time, placing the mixture in the ring caused intensive water loss, while the start of setting began at 54 min and the end was attained after 2 h and 3 min. This behavior of the clinker dust from the electrostatic precipitators was caused by over-burning, as before testing, the weighted specimen doze was sieved through a No. 09 sieve and the residual was 5% of the glassy grains. Based on the specimen’s weight, this was the reason for the observed reduction of solubility of the studied powder. It should also be noted that mixing with water led to an intense efflorescence as a result of the hydration of the alkali oxides in the clinker dust.

Thus, the binder based on the aspiration dust was the most effective as it obtained the maximum compressive strength of 50.7 MPa. Its low level of water absorption (11–12 wt. %) will make it possible to predict the high durability of structures built from this material (Figure 10).

The obtained test results provide an understanding of the complexity of the setting process in the binding system “mineral powder-alkali solution Na_2_SiO_3_”. In all of the examined systems, the highest reactivity was demonstrated by the alkali mixing fluid, which is responsible for curing cementitious binder due to its contact with anhydrous carbonic acid, in accordance with Equation (1):Na_2_SiO_3_ + CO_2_ + 2H_2_O = Si(OH)_4_ + Na_2_CO_3_(1)

At the same time, in the prepared systems with the mixed alkali, the addition of mineral powders contributes to the processes of structural formation, which has been confirmed by studies using scanning electron microscopy and X-ray diffraction analysis.

Compared with traditional concrete on cement CEM I 42.5 N, it was noted that, with nearly equivalent strength properties achieved, the standard consistency is 2–3 times higher, which is not a negative factor due to the absence of water, and hence, cracks. Reducing the density by 30% allows for the use of alkali-activated materials for lightweight and high-strength structures. The several-times-faster start and end of the setting time indicates an acceleration of construction production in the case of the use of AAMs. The water absorption by weight of the developed materials and of traditional concretes is nearly the same.

The importance of this study lies in the creation of new building materials with unique characteristics and the simultaneous disposal of cement production waste. This contributes to cost reductions and aims at environmental care. The areas of application of the materials follows from the characteristics obtained, and, in particular, their low density and high strength make it possible to build tall and durable buildings and structures.

Further development of the research topic may be associated with the development of new constructive technological solutions for the repair and restoration of buildings and structures. This will make it possible to reduce the cost of concrete repair work and reinforced concrete elements, as well as to expand and improve the formulations and technologies used for obtaining clinker-free binders in substandard and technogenic raw materials.

## 4. Conclusions

A comprehensive study of the recycling of cement industry waste for alkali-activated materials production was carried out. The tasks for achieving this goal were to characterize the cement production waste as a new binder and to comprehensively research the microstructure, fresh, physical, and mechanical properties of the alkali-activated material based on the binder. The following main conclusions were drawn, emphasizing the scientific novelty and practical significance of the work:Grains of cement production waste are represented by coarse volumetric particles with pronounced cleavage, small leaves, and the clear presence of minerals. The mineral composition of cement production waste is characterized by calcium silicates, which guarantee good binding properties.The results of the X-ray diffraction analysis of the samples based on the cement-free binder of alkaline activation using clinker dust and aspiration dust confirmed the presence of calcite, quartz, feldspar close to albite, micas, and zeolites. The obtained products of the chemical interaction of the components of the binder confirm the effectiveness of the newly developed AAM.As a result of comparing several binders, it was found that the binder based on aspiration dust with Na_2_SiO_3_ and Na_2_SiF_6_ is the most effective since its specimens achieved a density of 1.8 g/cm^3^, maximum compressive strength of 50.7 MPa, flexural strength of 5.6 MPa, and minimum setting time (starting at 24 min and ending at 36 min), with a water absorption of 12.8 wt. %.Compared with traditional concrete on cement CEM I 42.5 N, it was noted that with nearly equivalent strength properties achieved, the standard consistency is 2–3 times higher, which is not a negative factor due to the absence of water, and, hence, cracks. Reducing the density by 30% allows for the use of alkali-activated materials for lightweight and high-strength structures. Its several-times-faster setting start and end times indicate a potential acceleration of construction production. The water absorption by weight of the developed materials and traditional concretes is nearly the same.The research results will be of interest to specialists in the construction industry since the proposed recipes for eco-friendly alkali-activated materials are an alternative to expensive and energy-intensive Portland cement, and they enable the creation of strong and durable concrete and reinforced concrete composites.

## Figures and Tables

**Figure 1 materials-15-06660-f001:**
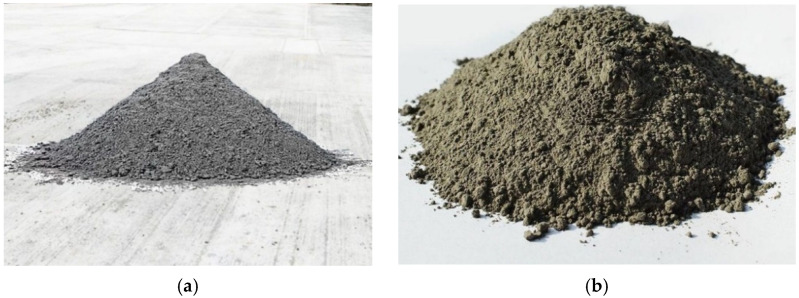
Appearance of the dusts: (**a**) clinker, and (**b**) aspiration.

**Figure 2 materials-15-06660-f002:**
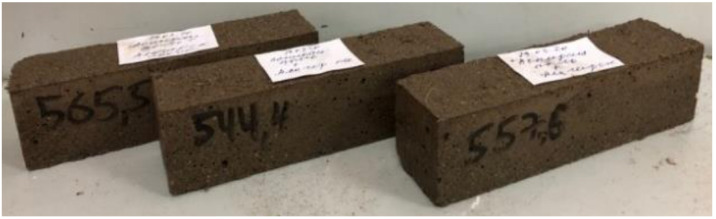
Sample beams with a size of 40 × 40 × 160 mm.

**Figure 3 materials-15-06660-f003:**
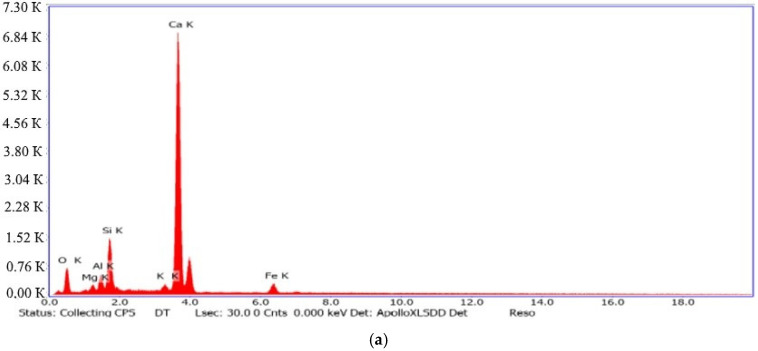
EDS spectra phases of the dusts: (**a**) clinker, and (**b**) aspiration.

**Figure 4 materials-15-06660-f004:**
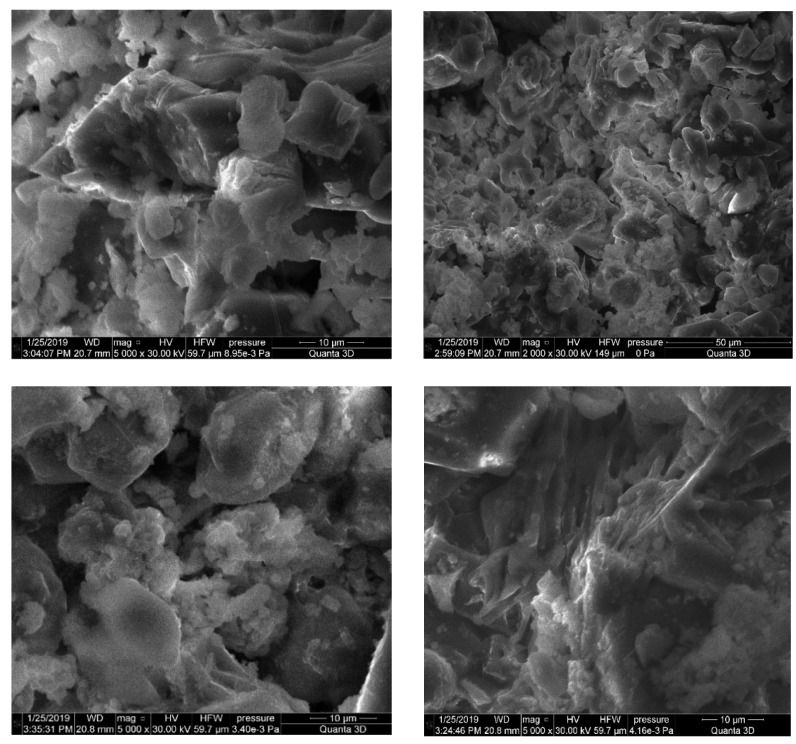
SEM images of the clinker dust.

**Figure 5 materials-15-06660-f005:**
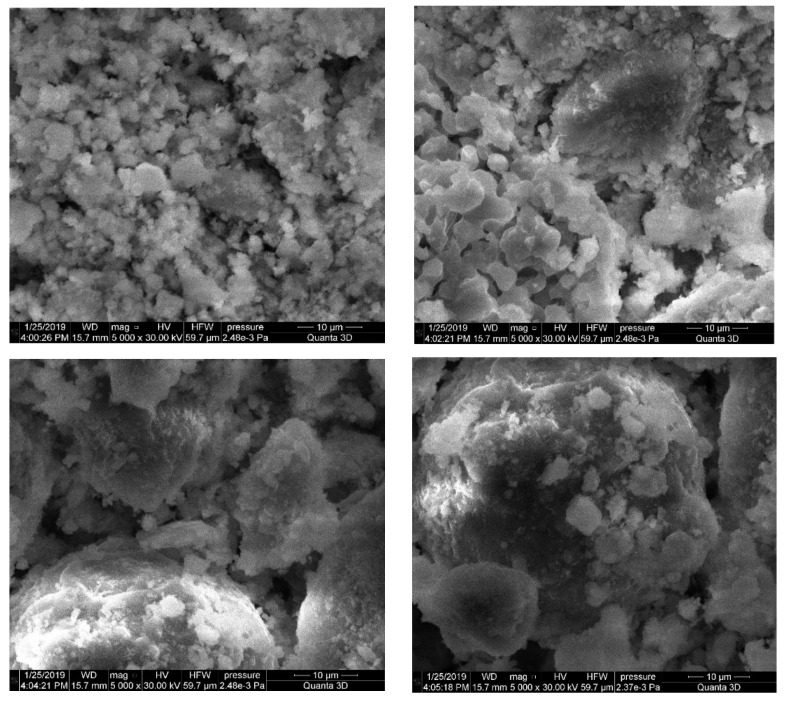
SEM images of the aspiration dust.

**Figure 6 materials-15-06660-f006:**
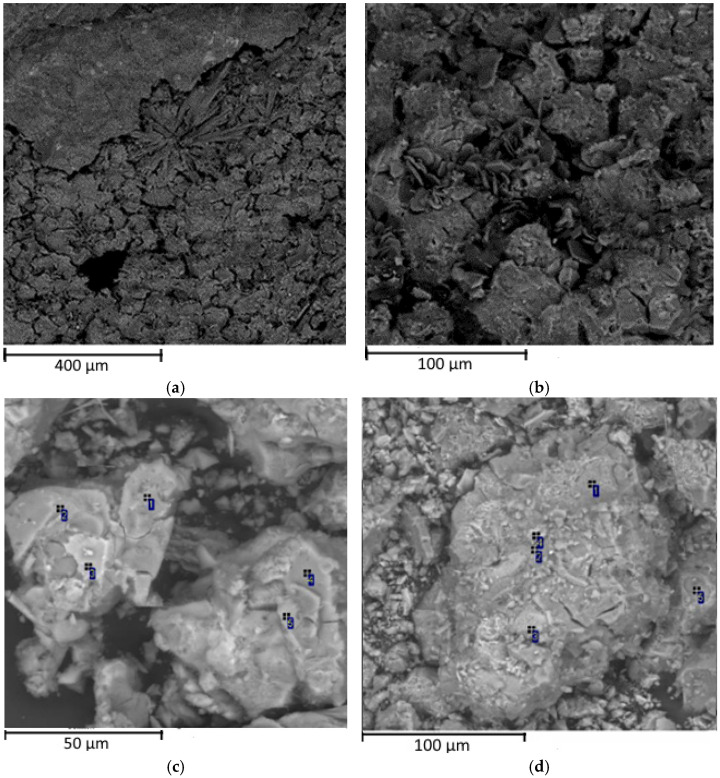
Microstructures of the AAMs with aspiration dust: (**a**) needle crystals, (**b**) lamellar crystals, (**c**,**d**) clusters.

**Figure 7 materials-15-06660-f007:**
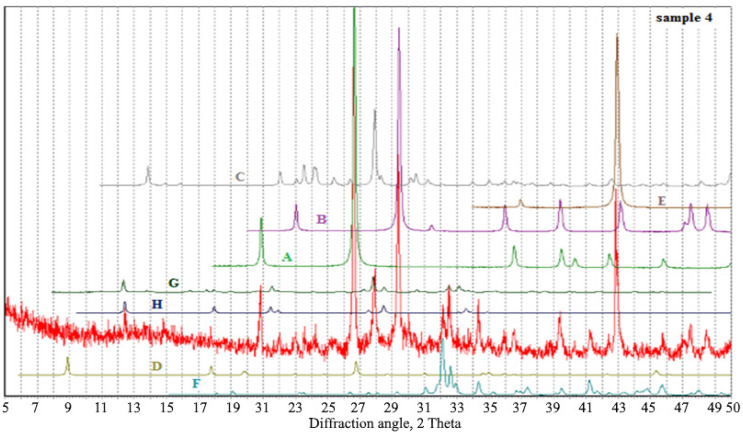
Diffraction pattern of the sample based on the aspiration dust from the electrostatic precipitators. The crystal phases for comparison comprise: A—quartz, B—calcite, C—albite, D—muscovite, E—magnesium oxide, F—larnite, G—analcime, and H—garronite.

**Figure 8 materials-15-06660-f008:**
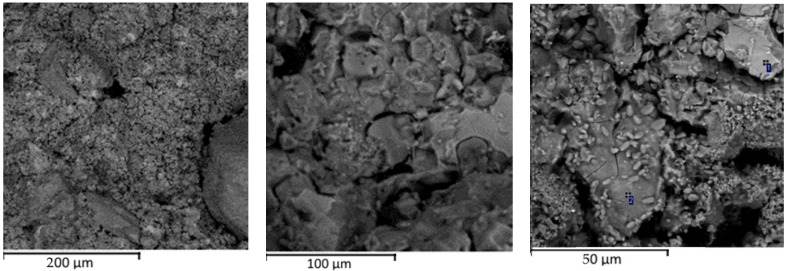
Microstructures of the AAMs with clinker dust.

**Figure 9 materials-15-06660-f009:**
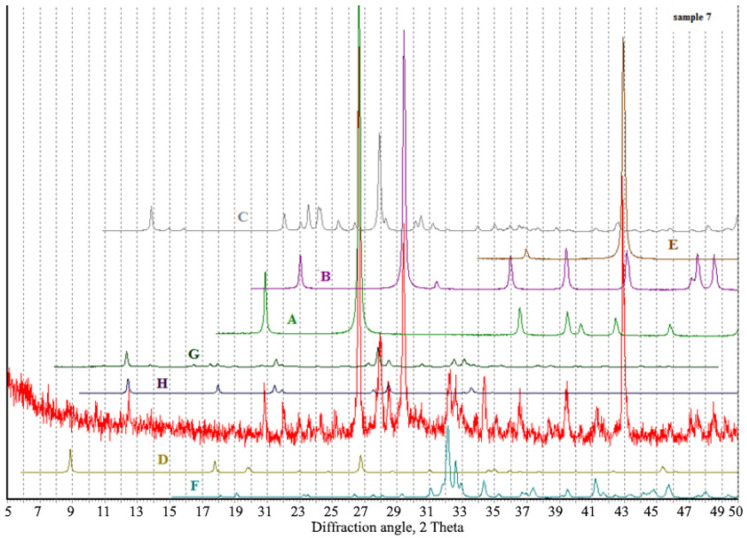
Diffraction pattern of the sample based on the clinker dust from the electrostatic precipitators. The crystal phases for comparison comprise: A—quartz, B—calcite, C—albite, D—muscovite, E—magnesium oxide, F—larnite, G—phillipsite, and H—garronite.

**Figure 10 materials-15-06660-f010:**
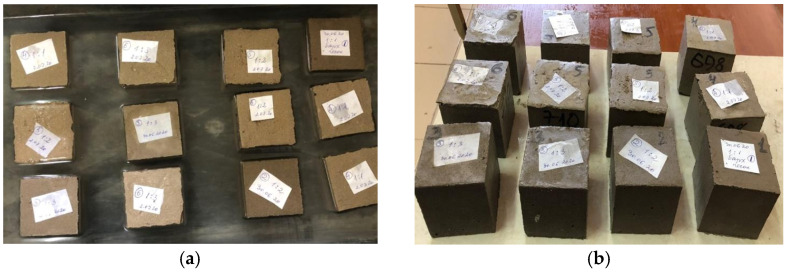
Water absorption test: (**a**) exposure to water, and (**b**) drying and weighing.

**Table 1 materials-15-06660-t001:** Physical properties of the dusts.

Electrostatic Precipitator Dust	True Density, kg/m^3^	Bulk Density, kg/m^3^	Specific Surface Area, m^2^/kg
Aspiration	2.59	1.13	280
Clinker	3.12	1.24	210

**Table 2 materials-15-06660-t002:** Chemical composition of the dusts (wt. %).

Electrostatic Precipitator Dust	CaO	SiO_2_	Al_2_O_3_	Fe_2_O_3_	K_2_O	MgO
Aspiration	64.14	20.31	4.68	3.47	6.43	0.97
Clinker	71.64	16.89	4.11	4.30	1.57	1.49

**Table 3 materials-15-06660-t003:** Mix proportions.

Mix ID	Components, kg per 1 m^3^						
	Clinker Dust	Aspiration Dust	Na_2_SiO_3_	H_2_O	Na_2_SiF_6_	NaOH	Sand
K1	690	-	270	-	-	69	1035
K2	690	-		270	-	69	1035
A1	-	690	270	-	-	69	1035
A2	-	690	-	270	-	69	1035
A3	-	690	270	-	16.8	69	1035

**Table 4 materials-15-06660-t004:** Results of the typical microphases analysis (images of the analysis are in Figure 6c,d).

Spectra	Na_2_O	MgO	Al_2_O_3_	SiO_2_	K_2_O	CaO	Fe_2_O_3_	Total
1	0.00	0.81	5.34	41.40	0.00	36.64	0.68	85.36
2	0.25	1.72	6.04	36.98	0.00	35.43	1.02	81.44
3	0.33	1.14	0,96	28.78	0.18	64.20	0.85	96.44
4	0.00	0.00	2,94	29.77	0.00	46.01	0.00	78.72
5	0.17	0.39	4,70	34.97	0.29	36.16	0.70	77.39

**Table 5 materials-15-06660-t005:** Results of the typical microphases analysis (images of the analysis are in the Figure 8).

Spectra	Na_2_O	MgO	Al_2_O_3_	SiO_2_	K_2_O	CaO	MnO	Fe_2_O_3_	Total
1	0.02	0.75	0.61	25.96	0.08	67.81	0.00	0.48	95.70
2	0.15	0.60	4.75	43.95	0.03	38.60	0.26	0.44	88.78

**Table 6 materials-15-06660-t006:** Fresh, physical, and mechanical properties of the AAMs.

Properties	Clinker Dust, S _sp_ = 210 m^2^/kg		Aspiration Dust, S_sp_ = 280 m^2^/kg		
	Type of Temperer				
	Na_2_SiO_3_	H_2_O	Na_2_SiO_3_	H_2_O	Na_2_SiO_3_ + Na_2_SiF_6_
Standard consistency, %	50.0	30.0	72.5	42.0	70.0
Setting time start/end, hours-min	00–40 01–20	00–54 02–03	00–16 00–31	06–08 07–16	00–24 00–36
Average density, g/cm^3^	1.80	1.70	1.70	1.70	1.80
Water absorption, wt. %	11.3	12.5	12.0	12.5	12.8
Strength, MPa: flexural compressive	4.8 43.7	0.3 7.8	5.1 50.1	0.3 7.9	5.2 50.7

## Data Availability

Not applicable.

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
