# Peer review of "Recycling of Cement Industry Waste for Alkali-Activated Materials Production"

_materials, 2022, doi:10.3390/ma15196660_

Round 1
Reviewer 1 Report
This manuscript investigated the structure formation of alkali-activated materials (AAM) based on aspiration dust and clinker dust. It is useful, but the paper's expression and analysis should be further improved. The detailed comments are as follows.
l Abstract should be refined, and the keywords are too many.
l The logic of the paragraphs should be reorganized. Why did the authors choose aspiration dust and clinker dust? What are the advantages of these two materials? The authors should be clarified clearly.
l The authors listed too much related research, which is hard to follow. Please make more analysis and summarization.
l The analysis of Section 3.1 is too simple. It is only a data report and lacks in-depth explanations.
l Section 3.3: the authors should compare the performance of alkali-activated materials with the traditional cement and explain the reason for the difference.
l Conclusions should be refined and list the most important conclusions.
l Figures 3, 7, and 9 should be replotted more clearly.
l Some typos are required to be corrected, such as Line 29, Line 214~221 fonts.
Author Response
Dear Reviewer 1!
Thank you for your interest in our manuscript. Your valuable comments helped make the manuscript even better. All corrections in the manuscript are highlighted in blue.
Comment 1: Abstract should be refined, and the keywords are too many.
Response: Abstract has been refined, and the keywords has been shortened.
Comment 2: The logic of the paragraphs should be reorganized. Why did the authors choose aspiration dust and clinker dust? What are the advantages of these two materials? The authors should be clarified clearly.
Response: At the end of the first page it says: «…clinker dust settles on the roofs of industrial buildings, then hydrates, forming buildups that can lead to the collapse of structures. Part of the dust (aspiration one) is captured by filters and needs to be further disposed of. Therefore, there is a need to minimize the consumption of cement with simultaneous utilization in the production of building materials.
Comment 3: The authors listed too much related research, which is hard to follow. Please make more analysis and summarization.
Response: All of these studies logically lead us to the choice and justification of the purpose of the article. Along with this, analysis and generalization are carried out throughout the introduction.
Comment 4: The analysis of Section 3.1 is too simple. It is only a data report and lacks in-depth explanations.
Response: In-depth explanations have been added
Comment 5: Section 3.3: the authors should compare the performance of alkali-activated materials with the traditional cement and explain the reason for the difference.
Response: It has been added: «Comparing with traditional concrete on cement CEM I 42.5 N, it has noted that with practically equivalent strength properties achieved, the standard consistency is 2-3 times higher, which is not a negative factor due to the absence of water, and hence cracks. The density reduced by 30% allows the use of alkali-activated materials for light-weight and high-strength structures. Several times faster start and end of setting time indicate the acceleration of construction production in the case of the use of AAM. Water absorption by weight of the developed materials and traditional concretes is almost the same.»
Comment 6: Conclusions should be refined and list the most important conclusions.
Response: Conclusions have been refined and listed the most important conclusions.
Comment 7: Figures 3, 7, and 9 should be replotted more clearly.
Response: Replotted
Comment 8: Some typos are required to be corrected, such as Line 29, Line 214~221 fonts.
Response: Corrected
Reviewer 2 Report
Dear authors,
The paper presents the results of the examination of cement industry wastes. The aim of this work is a comprehensive study of the structure formation of alkaline activated materials based on clinker dust and aspiration dust. The tasks to achieve this aim were the characterization of the cement production waste as a new binder and a comprehensive study of the microstructure, fresh, physical, and mechanical properties of alkali-activated material based on the binder.
The research results will be of interest to specialists in the construction industry since the proposed recipes for eco-friendly alkali-activated materials are an alternative to expensive and energy-intensive Portland cement and provide the creation of strong and durable concrete and reinforced concrete composites. The research is useful from a practical point of view, but there are many lacks that make it slightly suitable as a scientific work, even though scientific methods were used to characterize the starting raw materials, as well as alkali-activated materials.
Below are my remarks that you should pay attention to and correct the manuscript accordingly.
1. Figure 3 would be clearer if the elementary analysis were presented in a table.
2. Figures 4 and 5 it is necessary to be described in more detail.
3. Figure 7 should be displayed in a different way. This image can be used as an attachment, that is, if you want to deposit data.
4. As for Fig 9, I think it is the same as Fig 7 and should be displayed using another program.
5. In any case, apart from these corrections, it should be emphasized what is new in the work. In addition, the work must be strengthened in a scientific sense through the discussion of the results, and not just by stating the obtained data.
Sincerely
Author Response
Dear Reviewer 2!
Thank you for your interest in our manuscript. Your valuable comments helped make the manuscript even better. All corrections in the manuscript are highlighted in blue.
Comment 1: Figure 3 would be clearer if the elementary analysis were presented in a table.
Response: The chemical composition of the cement industry waste is given in Table.
Comment 2: Figures 4 and 5 it is necessary to be described in more detail.
Response: Figures 4 and 5 have been described in more detail.
Comment 3: Figure 7 should be displayed in a different way. This image can be used as an attachment, that is, if you want to deposit data.
Response: The XRD pattern is given in the traditional style, which makes it possible to determine the chemical composition of AAM
Comment 4: As for Fig 9, I think it is the same as Fig 7 and should be displayed using another program.
Response: The XRD pattern is given in the traditional style, which makes it possible to determine the chemical composition of AAM
Comment 5: In any case, apart from these corrections, it should be emphasized what is new in the work. In addition, the work must be strengthened in a scientific sense through the discussion of the results, and not just by stating the obtained data.
Response: The paper originality lies in the fact that for the first time a comprehensive study of the structure formation of alkali-activated materials (AAM) based on aspiration dust and clinker dust was carried out.
The discussion was expanded as the results were described
Reviewer 3 Report
The article is about recycling of the cement industry wastes for the alkali-activated materials production. However, some issues must to be addressed:
- Abstract: Please start by expressing the aim of this paper, followed by the rest of the information. Typically, the abstract should provide a broad overview of the entire project, summarize the results, and present the implications of the research or what it adds to its field.
- The bibliographic foundation is important and well executed, however some new discussions should be inserted, authors should consider some works in the literature, such as: DOI 10.1088/1757-899X/374/1/012019 or DOI 10.3390/ma14112967.
- Lines 161-163: please provide more details - explanation is too general.
- Characterization of those raw materials by EDS looks totally inefficient; is better to use XRF analysis.
- SEM images from 2019 are useless; why 4 images? Which is scientific purpose to include 4 images for figure 4 and 5??
- Figures 6 and 8: please enhance the clarity!
- Figures 7 and 9: XRD analysis is compulsory, but the way that the authors are treating this analysis, have only the role to confuse any reader. Put in evidence all phases; some of them are missing!
- Tables 4 and 5: FeO??? It can be Fe2O3 or Fe3O4?! Which one or both in which proportion??
- Caption from figure 10 is about weighting: where are data?! Results from table 6 are not enough!
- The results are merely presented, not properly discussed. Please add explanations for the observed changes. Please give an extended discussion on the obtained results and correlate your findings with previous literature studies and prospective applications.
- More analysis and interpretation of the results should be added for a clearer understanding of observed experimental phenomena.
- The authors must to provide some details about importance of the research and their applicability.
- Please rewrite the conclusions in a more quantitative form and enhance the clarity of the conclusion section in order to highlight the results obtained.
- General check-up and correction of the English language is suggested. There are still some minor typos and grammatical errors.
The author needs to address the abovementioned points for the betterment of the manuscript.
Author Response
Dear Reviewer 3!
Thank you for your interest in our manuscript. Your valuable comments helped make the manuscript even better. All corrections in the manuscript are highlighted in blue.
Comment 1: Abstract: Please start by expressing the aim of this paper, followed by the rest of the information. Typically, the abstract should provide a broad overview of the entire project, summarize the results, and present the implications of the research or what it adds to its field.
Response: Abstract has been refined
Comment 2: The bibliographic foundation is important and well executed, however some new discussions should be inserted, authors should consider some works in the literature, such as: DOI 10.1088/1757-899X/374/1/012019 or DOI 10.3390/ma14112967.
Response: Both articles have been carefully read and discussions on them have been inserted into the text.
Comment 3: Lines 161-163: please provide more details - explanation is too general.
Response: Explanation has been expanded
Comment 4: Characterization of those raw materials by EDS looks totally inefficient; is better to use XRF analysis.
Response: EDS is a special case of XRF based on the analysis of the emission energy of the X-ray spectrum, and this principle is used in most X-ray fluorescence spectrometers on the market.
Comment 5: SEM images from 2019 are useless; why 4 images? Which is scientific purpose to include 4 images for figure 4 and 5??
Response: The scientific purpose of these SEM images at one magnification is to study the diversity of morphological forms of this raw material.
Comment 6: Figures 6 and 8: please enhance the clarity!
Response: Enhanced
Comment 7: Figures 7 and 9: XRD analysis is compulsory, but the way that the authors are treating this analysis, have only the role to confuse any reader. Put in evidence all phases; some of them are missing!
Response: The description of XRD patterns was significantly expanded by him, arranged in a logical sequence
Comment 8: Tables 4 and 5: FeO??? It can be Fe2O3 or Fe3O4?! Which one or both in which proportion??
Response: Corrected to Fe2O3
Comment 9: Caption from figure 10 is about weighting: where are data?! Results from table 6 are not enough!
Response: Immediately before this figure, it is written that water absorption values ​​by mass were obtained at the level of 11-12%, which corresponds to an increase in the weight of the samples by this value.
During the revision, a paragraph was added comparing all measured characteristics of developed materials with similar ones of the developed materials
Comment 10: The results are merely presented, not properly discussed. Please add explanations for the observed changes. Please give an extended discussion on the obtained results and correlate your findings with previous literature studies and prospective applications.
Response: A discussion of the results has been added as appropriate, incl. added explanations for observed changes. An extended discussion of the results obtained has been given and the conclusions have been compared with previous literature studies and proposed applications.
Comment 11: More analysis and interpretation of the results should be added for a clearer understanding of observed experimental phenomena.
Response: A discussion of the results has been added as appropriate, incl. added explanations for observed changes. An extended discussion of the results obtained has been given and the conclusions have been compared with previous literature studies and proposed applications.
Comment 12: The authors must to provide some details about importance of the research and their applicability.
Response: Added: «The importance of this study lies in the creation of new building materials with unique characteristics with the simultaneous disposal of cement production waste. This contributes to cost reduction and aims at environmental care. The areas of application of materials follow from the characteristics obtained, in particular, low density and high strength make it possible to build high and durable buildings and structures.»
Comment 13: Please rewrite the conclusions in a more quantitative form and enhance the clarity of the conclusion section in order to highlight the results obtained.
Response: Conclusions have been refined and listed the most important conclusions
Comment 14: General check-up and correction of the English language is suggested. There are still some minor typos and grammatical errors.
Response: English has been verified by a native speaker
Round 2
Reviewer 2 Report
Dear authors,
I only have a few objections in your revised manuscript. You have to look at refs 4 and 5 in your manuscript and in this way you should to show your XRD results. If you do not do this, then it is necessary that Figs 4 and 5 are uniform. Title of the x axis of Fig 4 and 5 is not visible. Also, it is necessary to check the list of references and correct the following refs based on the instructions for authors: 6, 7, 24, 25, 26, 33, 34, 41, 42Also, it is necessary to check the list of references and correct the following refs based on the instructions for authors: 6, 7, 24, 25, 26, 33, 34, 41, 42.
Sincerely
Author Response
Figures 4 and 5 present SEM images of clinker dust and aspiration dust, respectively. And Figure 3 shows the EDS results for this raw material. All ref you indicated have been carefully edited.
Reviewer 3 Report
The article is suitable for publication.
Author Response
Thank you for appreciating our manuscript